# Immunohistochemical Distribution and Neurochemical Characterization of Huntingtin-Associated Protein 1 Immunoreactive Neurons in the Adult Mouse Lingual Ganglia

**DOI:** 10.3390/brainsci13020258

**Published:** 2023-02-03

**Authors:** Md Nabiul Islam, Yoshinori Sakurai, Yurie Hiwaki, Abu Md Mamun Tarif, Marya Afrin, Mirza Mienur Meher, Kanako Nozaki, Koh Hei Masumoto, Akie Yanai, Mir Rubayet Jahan, Koh Shinoda

**Affiliations:** 1Division of Neuroanatomy, Department of Neuroscience, Yamaguchi University Graduate School of Medicine, 1-1-1 Minami-Kogushi, Ube 755-8505, Japan; 2Department of Anatomy and Histology, Faculty of Veterinary Science, Bangladesh Agricultural University, Mymensingh 2202, Bangladesh; 3Department of Basic Laboratory Sciences, Faculty of Medicine and Health Sciences, Yamaguchi University Graduate School of Medicine, 1-1-1 Minami-Kogushi, Ube 755-8505, Japan; 4Department of Microbiology and Public Health, Faculty of Veterinary Medicine and Animal Science, Bangabandhu Sheikh Mujibur Rahman Agricultural University, Gazipur 1706, Bangladesh

**Keywords:** huntingtin-associated protein 1, lingual ganglia, neurodegeneration, neuroprotection

## Abstract

Huntingtin-associated protein 1 (HAP1) is a determinant marker for the stigmoid body (STB), a neurocytoplasmic physiological inclusion. STB/HAP1 enriched areas in the brain/spinal cord are usually protected from neurodegenerative diseases, whereas the regions with tiny amounts or no STB/HAP1 are affected. In addition to the brain/spinal cord, HAP1 is highly expressed in the myenteric/submucosal plexuses of the enteric nervous system in the gastrointestinal tract. The tongue is attached to the pharynx by the hyoid bone as an extension of the gastrointestinal system. To date, the immunohistochemical distribution and neurochemical characterization of HAP1 have not been elucidated in the lingual ganglia. Using immunohistochemistry and light microscopy, our current study demonstrates the expression and immunohistochemical phenotype of HAP1 in the lingual ganglia of adult mice. We showed that HAP1 was profoundly distributed in the intralingual ganglion (ILG) and the ganglia near the root of the tongue (which we coined as “lingual root ganglion”; LRG). Neurons in ILG and LRG exhibited high coexpression of HAP1 with NOS or ChAT. Furthermore, most HAP1-immunoreactive neurons contained SP, CGRP, and VIP immunoreactivity in both ILG and LRG. The current results might serve as an essential base for future studies to elucidate the pathological/physiological functions of HAP1 in the lingual ganglia.

## 1. Introduction

Huntingtin-associated protein 1 (HAP1) was first recognized for its binding to polyglutamine (polyQ)-expanded huntingtin, the causative protein for Huntington’s disease [1]. HAP1 is considered protective against cell death or apoptosis induced by huntingtin with an expanded polyQ sequence [2,3,4]. HAP1 is also known as an immunohistochemical marker of the stigmoid body (STB). STB is a neurocytoplasmic physiological inclusion expressed in different areas of the brain and spinal cord [5,6]. In the central nervous system (CNS) of normal rats or mice, HAP1 is profoundly distributed in the preoptic area, the bed nucleus of the stria terminalis, hypothalamus, amygdala, preganglionic sympathetic neurons, and the preganglionic parasympathetic neurons [7,8,9,10,11,12]. These HAP1-enriched regions in the CNS are generally protected from apoptosis in certain neurodegenerative disorders. Intriguingly, the areas in CNS with little or no HAP1 immunoreactivity, such as the striatum, neocortex, thalamus, cerebellum, and motor neurons, are primary targets in different neurodegenerative disorders [7,9,11]. This evidence suggests that STB/HAP1 has protective functions against neurodegeneration. In addition to Huntington’s disease, previous studies indicated that STB/HAP1 could also have protective effects against neurodegeneration in spinal and bulbar muscular atrophy [13], spinocerebellar ataxia type 17 [14], Machado-Joseph disease [15], and Joubert syndrome [16].

Evidence suggests that neurodegenerative diseases impact the enteric nervous system (ENS) in addition to the CNS [17]. ENS comprises the myenteric and submucosal plexuses, which reside within the entire gut wall [18]. Intriguingly, our recent study examined the distribution of HAP1 in the ENS throughout the gastrointestinal tract. HAP1 is intensely distributed in the submucosal and myenteric plexuses of ENS [19,20]. HAP1 is profoundly expressed in inhibitory motor neurons, interneurons, and excitatory motor neurons but is lacking in sensory neurons in the myenteric plexus. It is also abundantly expressed in all secretomotor and vasodilator neurons of Meissner’s plexuses. These indicate that the absence of HAP1 protectivity may cause the sensory neurons to be more vulnerable to cell death/apoptosis than HAP1-immunoreactive vasodilator/secretomotor neurons in Meissner’s plexuses and Dogiel type I neurons in myenteric plexus [19,20].

The tongue is connected to the pharynx by the hyoid bone and is an extension of the gastrointestinal system. The tongue comprises intrinsic (inferior longitudinal, superior longitudinal, transverse, and vertical) and extrinsic (genioglossus, styloglossus, hyoglossus, and palatoglossus) muscles. It is well-known that the tongue muscles are innervated by several cranial nerves [21,22]. Motor innervation for all extrinsic and intrinsic muscles is supplied via the hypoglossal nerve (CN XII). However, the palatoglossus muscle has vagal innervation (CN X). For sensory innervation, in the anterior two-thirds of the tongue, the general sensation is supplied by the lingual nerve (part of the trigeminal nerve, CNV). In contrast, taste sensation in the anterior two-thirds is provided via the chorda tympani branch of the facial nerve (CN VII). Furthermore, the touch and taste sensation in the posterior one-third is supplied by the glossopharyngeal nerve (CN IX). Somatosensory sensations are mediated via parasympathetic and sensory nerve fibers. 

Other than cranial innervation, the intrinsic neurons may also play a vital role in tongue functions [23]. The presence of intralingual ganglion (ILG) has been mentioned in previous studies [24,25]. ILG neurons lie along the muscles, glands, or near the circumvallate papillae [24]. Previous studies showed that ILG neurons expressed different neuropeptides, such as choline acetyltransferase (ChAT), vasoactive intestinal peptide (VIP), calcitonin gene-related peptide (CGRP), substance P (SP), and calcium-binding protein [26,27,28]. Our recent study demonstrated that HAP1 immunoreactivity is present in the cranial nerve preganglionic parasympathetic nuclei in the brainstem that innervate the tongue [11]. In addition, HAP1 is also highly expressed in the neurons of the myenteric and submucosal ganglion that contain ChAT, NOS, SP, VIP, Calbindin, or calretinin immunoreactivity [19,20]. To date, the immunohistochemical expression, neuroanatomical distribution, and neurochemical characterization of HAP1 in the tongue have not been studied. In the present study, we aim to investigate the immunohistochemical expression and neuroanatomical distribution of HAP1 in the lingual ganglia of the adult mouse. We also aim to elucidate the immunohistochemical phenotypes of HAP1 in the lingual ganglia.

## 2. Materials and Methods

### 2.1. Experimental Animals

Adult C57BL/6J were bred and housed at the animal facility of the Faculty of Medicine, Yamaguchi University, Japan. The mice were maintained in standard conditions (22 ± 2 °C, 12-h light/dark cycle,) with food, water, and nesting martial available ad libitum. For the current experiments, adult male mice were used. 

### 2.2. Characterization of Primary Antibodies 

The primary antibodies used in the current study are listed in Table 1. Characterization of HAP1 and other primary antibodies was performed in our previous studies or in previous works by other researchers (Table 1). All primary antibodies were available commercially. 

### 2.3. Tissue Preparations for Immunohistochemistry

Anesthetized (using sodium pentobarbital, 80 mg/kg, intraperitoneally) adult mice were transcardially perfused with PBS before introducing 4% paraformaldehyde in 0.1 M phosphate buffer (pH 7.4). Then, the tongue was dissected and post-fixed in 4% paraformaldehyde overnight at 4 °C. The tongues were kept in 30% sucrose solution for 5–6 d. The tissue was mounted in a cryostat embedding medium and snap-frozen in powdered dry ice. The tongues were sectioned using a cryostat (thickness of 30 µm) and kept at 4 °C in PBS for storage until used for immunohistochemistry. For the myenteric plexus, the whole mount of the duodenum was prepared as described in our previous study [19].

### 2.4. Single-Label Immunohistochemistry

Immunoperoxidase staining was performed as narrated in our earlier reports [8,9,10,11,30,31,32]. In brief, tongue sections (30 µm of thickness) were rinsed three times (for 10 min each) in PBS and then treated with 10% donkey serum and 0.3% TritonX-100 in PBS for 2 h at 4 °C. Next, sections were pretreated with 1.5% hydrogen peroxide and 50% methanol mixture for 30 min at 4 °C. Primary antibody for HAP1 was applied in 1% donkey serum and 0.3% TritonX-100 in PBS for 5 d at 20 °C. A blocking peptide against the HAP1 primary antibody was preincubated with the primary antibody at 4 °C overnight for the preadsorption test. After washing in PBS, the tongue sections were incubated with a secondary antibody (AP180B, donkey anti-goat secondary antibody, Millipore, Burlington, MA, USA) for 2 h at 20 °C. After being incubated with streptavidin-biotin (Dako, Glostrup, Denmark, P0397) at 20 °C for 2 h, the sections were stained with a mixture of DAB (3,3′-DAB, Wako, Osaka, Japan) and nickel ammonium sulfate (Sigma Aldrich, Tokyo, Japan) solution in 0.05 M Tris-HCl buffer containing 1% hydrogen peroxide for 10–15 min. The tongue sections were then mounted on glass slides. Then sections were air-dried for 60 min and dehydrated with ascending grades of alcohol, and xylene. Finally, the sections were embedded with Entellan New.

### 2.5. Double-Label Immunohistochemistry 

Immunofluorescence staining was performed following our earlier studies [19,29]. In brief, tongue sections (or whole mount for duodenum) were incubated with 10% NDS containing 0.3% Triton X-100 at 20 °C for 3 h for blocking. After washing three times (for 10 min each in PBS) the sections were treated with a mixture of primary antibodies (goat anti-HAP1 with rabbit anti-ChAT, rabbit anti-NOS, rabbit anti-SP, rabbit anti-VIP, or rabbit anti-CGRP for lingual ganglia; or goat anti-ChAT with rabbit anti-NOS for myenteric plexus) in 1% NDS in PBS containing 0.3% Triton X-100 at 20 °C for 5 d. Then, a mixture of secondary antibodies (Alexa 594 donkey anti-goat, A11058 with Alexa 488 donkey anti-rabbit, A32790) diluted in 1% NDS in PBS for 2 h at 20 °C, was applied to the sections. After washing three times in PBS (for 10 min each), the sections were mounted onto glass slides with a mounting medium. Finally, the sections were air-dried for 60 min and embedded with Fluoromount Plus (Diagnostic Biosystems, Pleasanton, CA, USA; K048).

### 2.6. Imaging

Light microscopic photomicrographs were taken with a Nikon Eclipse E80i photomicroscope (Nikon Instruments, Tokyo, Japan) attached to a Lumenera USB 2.0 camera (Lumenera Corporation, Ottawa, ON, Canada). Confocal photomicrographs were taken with an LSM510 confocal laser scanning microscope (Carl Zeiss, Germany). Single optical sections were captured (1024 × 1024 pixels). Minimal bright/contrast adjustments and plates of photomicrographs were performed with Photoshop 2021 (Adobe, San Jose, CA, USA). 

### 2.7. Cell Counting 

For cell counting, 20× objective immunofluorescence images were taken and transferred into ImageJ software, version 1.51i (NIH, Bethesda, MD, USA). Coexpression ratios for HAP1/ChAT, ChAT/HAP1, HAP1/NOS, NOS/HAP1, HAP1/SP, SP/HAP1, HAP1/CGRP, CGRP/HAP1, HAP1/VIP, VIP/HAP1, ChAT/NOS, NOS/ChAT were calculated from the total number of HAP1, ChAT, NOS, SP, VIP, or CGRP-ir cells, as well as those double immunostained for HAP1 with ChAT, NOS, SP, VIP, CGRP, and ChAT with NOS [11,33]. Six sections (every alternative section) were used from one animal. For cell counting, data from six animals were used in our current study.

## 3. Results

### 3.1. Immunohistochemical Expression and Distribution of HAP1 in the Lingual Ganglia

HAP1 immunoreactivity was detected in the ILG cells scattered mostly in the anterior one-third of the tongue (Figure 1A). However, a few HAP1-ir ILG cells were also present in the intermediate and posterior one-third of the mouse tongue (Figure 1A). Intriguingly, in addition to the ILG, a distinct population of HAP1-ir ganglionic cells was found at the root of the tongue (Figure 1A). To the best of our knowledge, the presence of lingual ganglia at the root of the tongue has not been reported previously. Hence, in our present study, we coined the name of the HAP1-ir ganglia at the root of the tongue as the “lingual root ganglion (LRG)”. Most of the HAP1-ir cells in ILG or LRG showed cytoplasmic diffuse immunoreactivity in which the STBs were not clearly found (Figure 1B,C). In ILG, one ganglion contained 5–8 HAP1-ir cells, whereas in LRG, one ganglion contained 10–22 HAP1-ir cells (Figure 1B,C). In the preadsorption test, the immunoreaction for HAP1 was abolished in the ILG and LRG (Figure 1D). Double immunofluorescence histochemistry for HAP1 and nuclear neuronal marker (NeuN) showed that most HAP1-ir cells had clear NeuN immunoreaction in their nuclei in ILG (Figure 1E–G) or LRG (Figure 1H–J), confirming that the cells that were immunoreactive for HAP1 in the lingual ganglia showed the attributes of the neuron.

### 3.2. Immunohistochemical Relationships of HAP1 with the Markers of Parasympathetic Neurons

Based on histochemical studies, the ganglion cells in the tongue have been considered to have parasympathetic functions [34]. To elucidate the immunohistochemical associations between HAP1 and parasympathetic autonomic neurons in the lingual ganglia, double immunofluorescence staining for HAP1 with ChAT and NOS was performed. HAP1-ir neurons were highly coexpressed with ChAT-ir or NOS-ir neurons in ILG (Figure 2A–F) or LRG (Figure 2G–L). The results of cell counting showed that, the ILG, 84.36% of HAP1-ir neurons had immunoreactivity for ChAT, and 98.42% of the ChAT-ir neurons contained immunoreactivity for HAP1 (Table 2). At the same time, 90.91% of HAP1-ir neurons had immunoreactivity for NOS, and 99.37% of the NOS-ir neurons contained immunoreactivity for HAP1 in the ILG. The results of cell counting also showed that 99.39% of HAP1-ir neurons had immunoreactivity for ChAT, and 99.28% of the ChAT-ir neurons contained immunoreactivity for HAP1 in the LRG (Table 2). A total of 86.47% of HAP1-ir neurons had immunoreactivity for NOS, and 98.54% of the NOS-ir neurons contained immunoreactivity for HAP1in the LRG (Table 2).

These results were intriguing in that HAP1 was coexpressed with excitatory parasympathetic neuronal marker ChAT and inhibitory parasympathetic neuronal marker NOS. The excitatory parasympathetic and inhibitory neurons are usually different entities in the myenteric plexuses of ENS (Figure 3A–C). To answer whether the ChAT and NOS are expressed in the same neurons in the lingual ganglion, our current double-stained results for ChAT and NOS in lingual ganglia indicated that almost all the ChAT-ir neurons contained NOS immunoreactivity in the ILG (Figure 3D–F) or the LRG (Figure 3G–I). These results confirmed that, unlike the myenteric plexuses, ChAT and NOS are expressed in the same neurons in the lingual ganglion. 

### 3.3. Immunohistochemical Relationships of HAP1 with CGRP, SP, and VIP

It has been reported that neuropeptides, such as CGRP, SP, and VIP, are expressed in the lingual ganglionic cells. The functional significance of these neuropeptides in the lingual ganglia needs to be better clarified, but they are believed to modulate sensory functions in the tongue [25,28]. To examine the neurochemical relationships between HAP1 and plausible intrinsic sensory neurons in lingual ganglia, double immunofluorescence histochemical staining for HAP1 with CGRP, SP, and VIP was carried out (Figure 4). The current cell counting exhibited that 81.84% of HAP1-ir neurons had immunoreactivity for CGRP, and 85.67% of CGRP-ir neurons contained immunoreactivity for HAP1 in the ILG (Figure 4A–C, Table 2). A total of 99.68% of HAP1-ir neurons had immunoreactivity for SP, and 99.57% of the SP-ir neurons contained immunoreactivity for HAP1 in the ILG (Figure 4D–F, Table 2). In addition, 99.73% of HAP1-ir neurons had immunoreactivity for VIP, and 98.19% of the VIP-ir neurons contained immunoreactivity for HAP1 in the ILG (Figure 4G–I, Table 2). Results of cell counting also demonstrated that 81.69% of HAP1-ir neurons had immunoreactivity for CGRP, and 89.27% of the CGRP-ir neurons contained immunoreactivity for HAP1 in the LRG (Figure 4J–L, Table 2). A total of 98.74% of HAP1-ir neurons had immunoreactivity for SP, and 99.17% of the SP-ir neurons contained immunoreactivity for HAP1 in the LRG (Figure 4M–O, Table 2). In addition, 89.29% of HAP1-ir neurons had immunoreactivity for VIP, and 99.69% of the VIP-ir neurons contained immunoreactivity for HAP1 in the LRG (Figure 4P–R, Table 2). 

## 4. Discussion

The immunohistochemical expression and neuroanatomical distribution of HAP1 have been analyzed previously in the brain and spinal cord in CNS [7,8,9,12,16,31,35,36,37,38,39,40,41,42], in the dorsal root ganglion of the peripheral nervous system [10], and the submucosal or myenteric ganglia throughout the ENS [19,20]. In addition, HAP1 is distributed in the peripheral organs, including the pancreas, pituitary gland, and thyroid gland. Furthermore, HAP1 immunoreactivity is also present in enteroendocrine glands in the gastrointestinal mucosa [19,20,29]. However, its immunohistochemical expression, regional distribution, and neurochemical phenotypes in lingual ganglia have not been studied yet. By employing immunohistochemistry, our current study is the first to demonstrate the presence of HAP1 immunoreactivity in the neurons of the lingual ganglia. Our recent study is also the first to clarify the immunohistochemical relationships of HAP1 with ChAT, NOS, VIP, SP, and CGRP immunoreactive neurons in mouse lingual ganglia. ILG neurons are scattered along the muscles, glands, or near the circumvallate papillae [24]. Our current study found that HAP1 was highly expressed in the ILG throughout the tongue. In addition to ILG, we also found HAP1 immunoreactivity in the lingual ganglia scattered in the root of the tongue and coined the name of the ganglia as LRG. To the best of our knowledge, our current study is the first to report the presence of LRG in the tongue. Our current immunohistochemical results show that there are two types of lingual ganglia in the mouse tongue, ILG and LRG.

In both ILG and LRG, immunoreactivity for HAP1 was diffusely expressed in the cytoplasm of neurons. It is well-known that HAP1 is the determinant marker of STB. However, the STBs were not clearly detected in ILG or LRG in our current immunostaining. HAP1 has two isoforms, HAP1A and HAP1B [9,36,39,43]. HAP1A/HAP1B expression ratio is essential to form STBs [37,43] in the cytoplasm of the HAP1-contained neurons. The cells with intense HAP1B usually contain a low expression ratio for HAP1A/HAP1B. The comparatively smaller HAP1-ir STBs are usually concealed by diffusely expressing intense HAP1 immunoreactivity in the cytoplasm [11,37]. It is possible that more HAP1B than HAP1A is expressed in the mouse lingual ganglia. It has been reported that the ratios of HAP1A to HAP1B are also varied in different brain areas; more HAP1A than HAP1B is found in the spinal cord and olfactory bulb, whereas a similar amount of HAP1A and HAP1B is present in the hypothalamus or amygdala [1,2,8,9]. However, more HAP1B than HAP1A seems to be present in the dorsal root ganglion, submucosal, and myenteric ganglion in the peripheral nervous system [10,19,20] 

The tongue is innervated by sympathetic, parasympathetic, and sensory fibers via different pathways of cranial nerves. Motor innervation for all extrinsic and intrinsic muscles is provided via the CN X and CN XII. For sensory innervation, general sensation is supplied by the CNV (lingual nerve), and taste sensation is provided by CN VII and CN IX. Autonomic sensations are mediated via parasympathetic and sensory nerve fibers [21,22]. In addition to cranial nerve innervation, the presence of lingual ganglia has also been reported in mammalian and non-mammalian tongues. The lingual ganglionic cells are immunoreactive for ChAT, NOS, VIP, SP, NPY, and CGRP [25,26,44]. It has been reported that the ganglionic cells in the tongue modulate parasympathetic functions [34]. Furthermore, it is believed that the lingual ganglionic cells can also modulate sensory functions in the tongue [25,28]. The remarkable finding in the present study was that a high percentage of the ChAT, NOS, VIP, SP, or CGRP-ir neurons coexpressed with HAP1-ir neurons in ILG and LRG. Mounting evidence suggests that HAP1 can modulate stress response and feeding behaviors and play a vital role in early brain development [35,40,41,42]. It is intriguing to speculate that the high coexpression rate of HAP1 with different neurochemical markers in lingual ganglia probably reflects HAP1′s involvement in modulating intrinsic sensory and parasympathetic information under particular physiological conditions in the tongue. It has been known that HAP1 can interact with cytoskeletal proteins to modulate the intracellular trafficking of several proteins or to recycle/stabilize membrane receptors [45,46]. Intracellular membrane trafficking is a fundamental process essential for various neuronal functions. The high expression of HAP1 in ILG or LRG may suggest the involvement of HAP1 in regulating sensory or autonomic functions in the tongue by modulating intracellular membrane trafficking. However, detailed physiological and morphological experiments need to be conducted in the future to elucidate how HAP1 can regulate the autonomic (parasympathetic) and sensory activities in the tongue. 

It is believed that HAP1 has putative protective functions against apoptosis/cell death in stress conditions or certain neurodegenerative diseases. Abnormal interactions of mutant protein with HAP1 affect the intracellular transport of selected molecules, which is an essential contributor to the pathogenesis of neurodegenerative diseases. To prevent apoptosis-induced nuclear translocation, HAP1 can also sequester the casual mutant molecules of some neurodegenerative diseases and trap the toxic aggregation in the cytoplasm of the cells [14,15,16]. Intriguingly, brain or spinal cord regions with low/no HAP1 immunoreactivity are the main targets of certain neurodegenerative disorders. In contrast, the areas in the CNS with high HAP1 immunoreactivity are generally protected from apoptosis in neurodegenerative disorders [7,9,11]. In our current study, abundant HAP1 expression in ILG and LRG might suggest that HAP1-immunoreactivity could provide beneficial stability to the lingual ganglion in certain stress conditions. It is tempting to speculate that the tongue functions, such as feeding behavior in the fetal stage (swallowing of amniotic fluid), aging stage, or in diseased conditions (where cranial neurons are affected), might be protected by the presence of HAP1 in the lingual ganglia. However, future detailed studies will be needed that use HAP1 knock-out mice to clarify how HAP1 modulates these functions.

## 5. Conclusions

The present study is the first to elucidate the detailed neuroanatomical distribution and immunohistochemical characterization of HAP1 in the mouse lingual ganglia. Our current study is also the first to show the existence of a new lingual ganglion at the root of the mouse tongue, which we coined as LRG. HAP1 was highly expressed in both ILG and LRG. Our immunohistochemical results exhibit that HAP1-ir neurons are coexpressed with ChAT, NOS, SP, VIP, and CGRP-ir neurons. The present results might indicate the plausible functions of HAP1 in protecting or regulating neurons in ILG and LRG that are related to intrinsic parasympathetic autonomic and sensory functions in the tongue. The present study may serve as an important base for future studies that aim to elucidate the pathological or physiological functions of HAP1 in the lingual ganglia. The tongue has dual innervation, cranial innervation, and intrinsic intralingual innervation. Cranial innervation has been well studied previously [21,22], and our current results might clarify the intrinsic intralingual innervation of the tongue.

## Figures and Tables

**Figure 1 brainsci-13-00258-f001:**
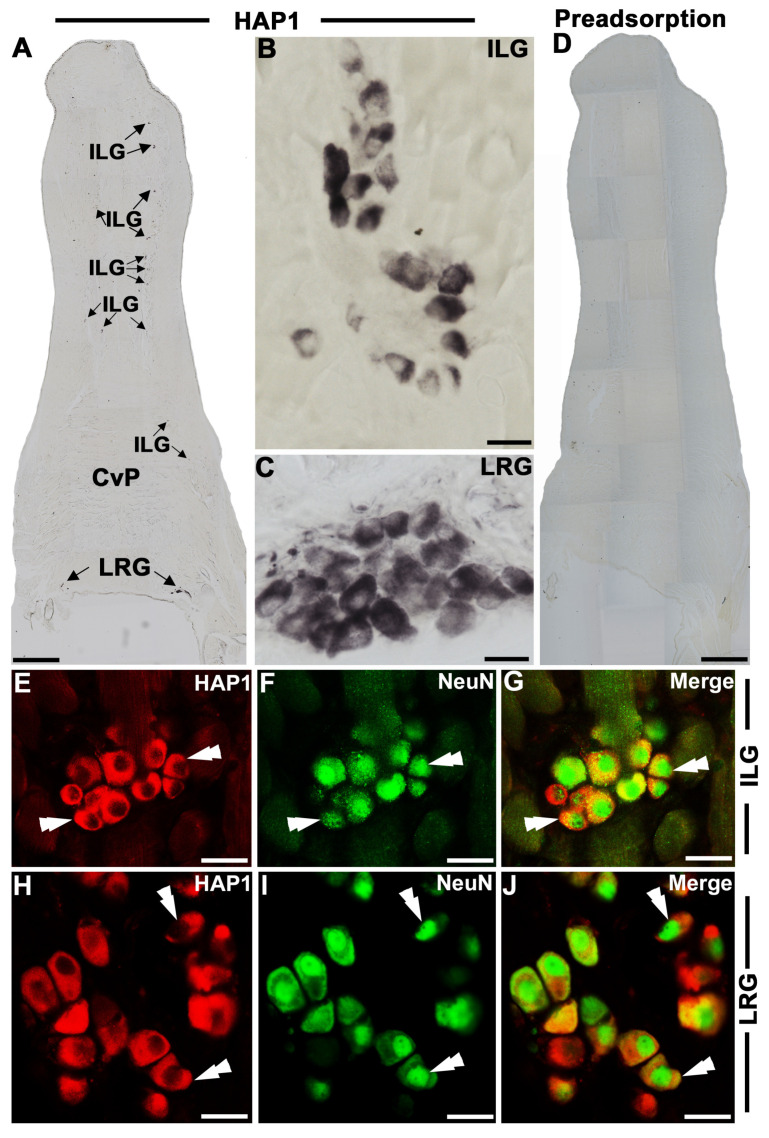
Immunohistochemistry for huntingtin-associated protein 1 (HAP1) in the lingual ganglia of the adult mouse. (**A**) Immunoperoxidase staining shows HAP1 immunoreactivity in the ILG and LRG. (**B**,**C**) enlargement of ILG and LRG indicating diffuse cytoplasmic HAP1-ir cells. (**D**) Preadsorption test (preincubation of HAP1 antibody with a blocking peptide) abolished the HAP1 immunoreactivity both in ILG and LRG. (**E**–**J**) Double immunofluorescence staining of HAP1 and NeuN in ILG and LRG. Cells positive for both HAP1 and NeuN are indicated by double arrowheads. ILG, intralingual ganglion; LRG, lingual root ganglion; CvP, circumvallate papillae. Scale bar = 400 µm in (**A**,**D**), 20 µm in (**B**,**C**), and 25 µm in (**E**–**J**).

**Figure 2 brainsci-13-00258-f002:**
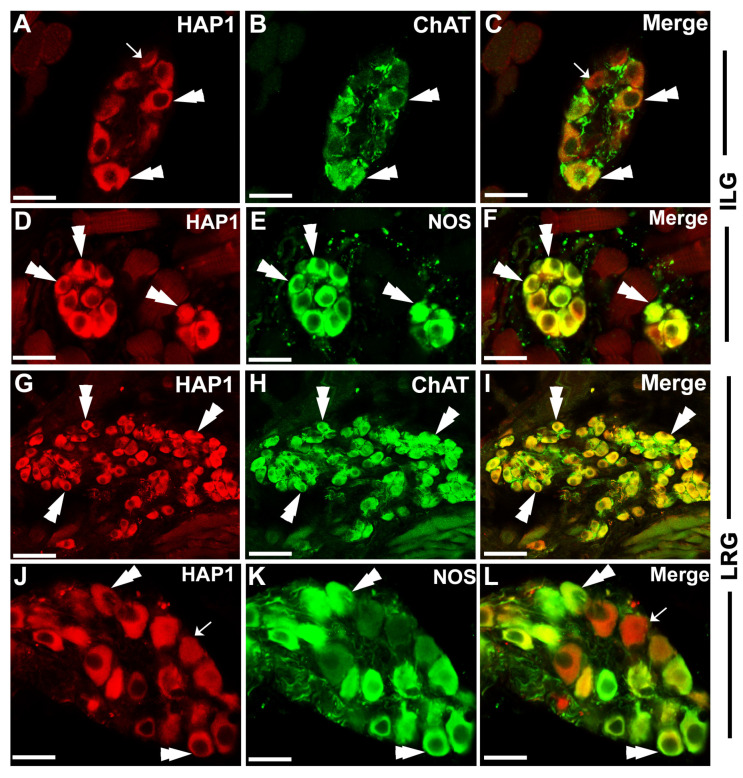
Double immunofluorescence staining for HAP1 with ChAT or NOS. Photomicrograph showing the coexpression of HAP1 with ChAT in ILG (**A**–**C**) or LRG (**G**–**I**); and HAP1 with NOS in ILG (**D**–**F**) or LRG (**J**–**L**). Cells positive for both HAP1 and NOS or ChAT are indicated by double arrowheads. Cells single-positive for HAP1 are indicated by arrows. ILG, intralingual ganglion; LRG, lingual root ganglion. Scale bar = 25 µm in (**A**–**F**,**J**–**L**), and 50 µm in (**G**–**I**).

**Figure 3 brainsci-13-00258-f003:**
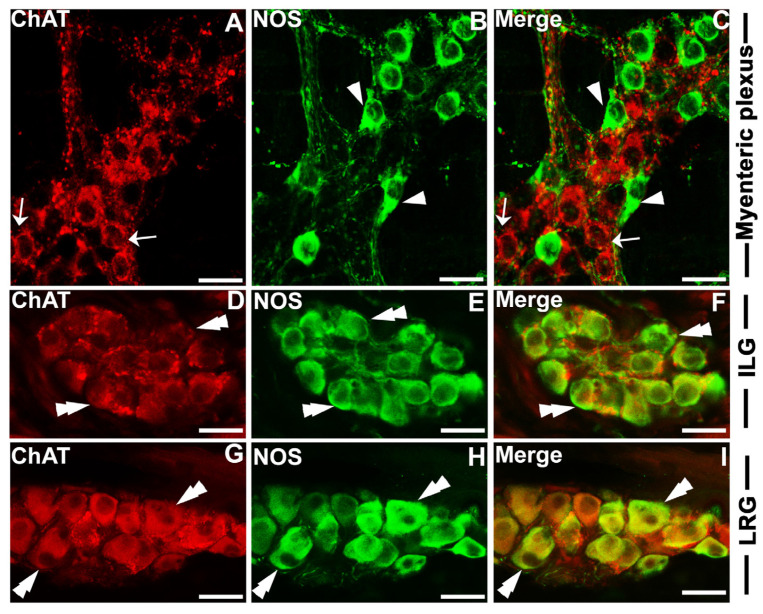
Double immunofluorescence histochemistry for ChAT with NOS in myenteric ganglia and lingual ganglia. Photomicrograph showing the coexpression of ChAT with NOS in the Myenteric plexus in the duodenum (**A**–**C**), ILG (**D**–**F**), and LRG (**G**–**I**). Cells single-positive for ChAT are indicated by arrows. Cells single-positive for NOS are indicated by arrowheads. Cells positive for both ChAT and NOS are indicated by double arrowheads. ILG, intralingual ganglion; LRG, lingual root ganglion. Scale bar = 25 µm in (**A**–**I**).

**Figure 4 brainsci-13-00258-f004:**
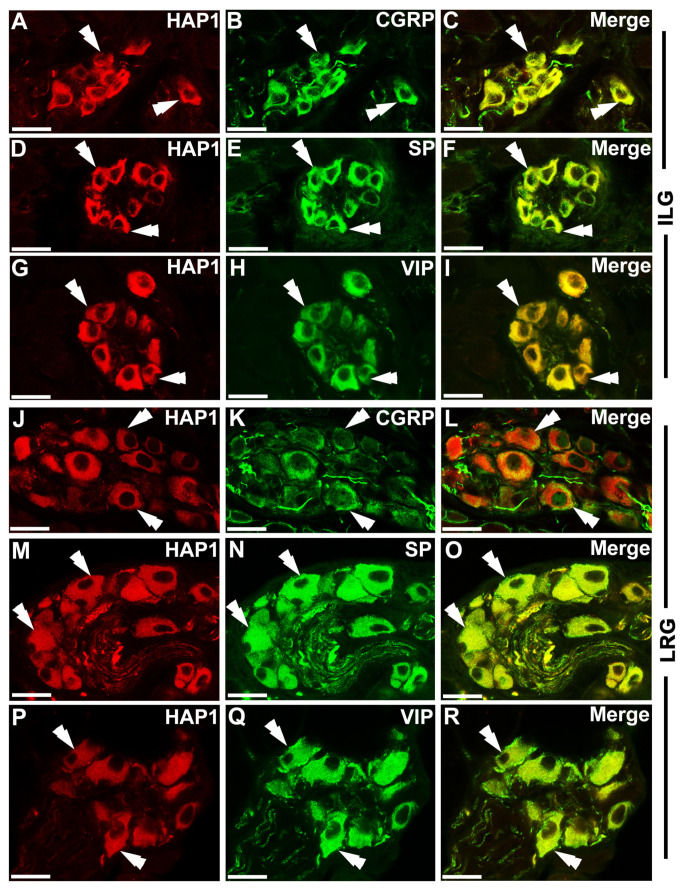
Double immunofluorescence histochemistry for HAP1 with CGRP, SP, or VIP. Photomicrograph showing the coexpression of HAP1 with CGRP in ILG (**A**–**C**) or LRG (**J**–**L**); HAP1 with SP in ILG (**D**–**F**), or LRG (**M**–**O**); and HAP1 with VIP in ILG (**G**–**I**), or LRG (**P**–**R**). Cells positive for both HAP1 and CGRP, HAP1 and SP, or HAP1 and VIP are indicated by double arrowheads. ILG, intralingual ganglion; LRG, lingual root ganglion. Scale bar = 25 µm in (**A**–**R**).

**Table 1 brainsci-13-00258-t001:** Details of primary antibodies used in the current study.

Name	Immunogen	Code	Source	Dilution	Reference
Goat Polyclonal anti-HAP1 (R19)	Rat HAP1 C-terminus	Cat# sc-8770, RRID: AB_647322	Santa Cruz Biotechnology, Santa Cruz, CA, USA	1:10,000	[29]
Rabbit Polyclonal anti-NeuN	Synthetic peptide of Human NeuN aa 1–100	Cat# ab177487, RRID: AB_2532109	Abcam, Cambridge, UK	1:5000	[10]
Rabbit Polyclonal anti-ChAT	Synthetic peptide within Pig Choline Acetyltransferase aa 150–250	Cat# ab178850, RRID: AB_2721842	Abcam, Cambridge, UK	1:5000	[11]
Goat Polyclonal anti-ChAT	Human placental ChAT	Cat# AB144P, RRID: AB_11214092	Millipore, Billerica, MA, USA	1:5000	[9]
Rabbit Polyclonal anti-NOS	C-terminus synthetic peptide of human nNOS coupled to KLH	Cat# 24287, RRID: AB_572256	Immunostar, Hudson, WI, USA	1:3000	[9]
Rabbit Polyclonal anti-SP	Synthetic SP coupled to KLH with carbodiimide	Cat# 20064, RRID: AB_572266	Immunostar, Hudson, WI, USA	1:2000	[10]
Rabbit Polyclonal anti-CGRP	CGRP-KLH (rat)	Cat# C8198, RRID: AB_259091	Sigma-Aldrich, St. Louis, MO, USA	1:2000	[20]
Rabbit Polyclonal anti-VIP	Porcine VIP coupled to bovine thyroglobulin with carbodiimide	Cat# 20077, RRID: AB_572270	Immunostar, Hudson, WI, USA	1:5000	[19]

ChAT, choline acetyltransferase; CGRP, calcitonin gene-related peptide; HAP1, Huntingtin-associated protein 1; NOS, nitric oxide synthase; NeuN, neuronal nuclei; VIP, Vasoactive Intestinal Peptide; SP, substance P.

**Table 2 brainsci-13-00258-t002:** Coexpression ratios of HAP1 with different neurochemical markers in the lingual ganglia.

Relationship of HAP1 with Different Markers (ratio %)		ILG	LRG
ChAT	ChAT/HAP1	84.36 ± 2.1	99.39 ± 0.2
	HAP1/ChAT	98.42 ± 3.2	99.28 ± 0.3
NOS	NOS/HAP1	90.91 ± 1.2	86.47 ± 1.9
	HAP1/NOS	99.37 ± 0.3	98.54 ± 0.7
CGRP	CGRP/HAP1	81.84 ± 3.2	81.69 ± 2.2
	HAP1/CGRP	85.67 ± 2.9	89.27 ± 1.7
SP	SP/HAP1	99.68 ± 0.4	98.74 ± 0.4
	HAP1/SP	99.57 ± 0.2	99.17 ± 0.3
VIP	VIP/HAP1	99.73 ± 0.3	89.29 ± 2.8
	HAP1/VIP	98.19 ± 0.2	99.69 ± 0.3

ChAT, choline acetyltransferase; CGRP, calcitonin gene-related peptide; HAP1, Huntingtin-associated protein 1; NOS, nitric oxide synthase; NeuN, neuronal nuclei; SP, substance P; VIP, Vasoactive Intestinal Peptide. Values represented the mean SEM (n = 6).

## Data Availability

The data presented in this study are available upon reasonable request from the corresponding author.

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
