# Peer review of "Immunohistochemical Distribution and Neurochemical Characterization of Huntingtin-Associated Protein 1 Immunoreactive Neurons in the Adult Mouse Lingual Ganglia"

_brainsci, 2023, doi:10.3390/brainsci13020258_

Round 1

Reviewer 1 Report

The Authors raise the interesting issue, considering the expression and distribution of HAP1 protein in lingual ganglia in mice which have never been studied before. It may have potential impact in studies on HAP1 under pathological conditions and its effect on the development/protection of the neurodegeneration disorders.

Minor comments for the text:

Abstract – ILG – the lack of shortcut extension.

Introduction – Line 38-39. The sentence about HAP1 is not fully understandable – assumption that HAP1 interact intentionally to protect against apoptosis. Moreover, polyglutamine domain exist both in normal and mutant huntingtin. It should be specified.

Author Response

The Authors raise the interesting issue, considering the expression and distribution of HAP1 protein in lingual ganglia in mice which have never been studied before. It may have potential impact in studies on HAP1 under pathological conditions and its effect on the development/protection of the neurodegeneration disorders.

We thank the reviewer for carefully reading our manuscript and for the valuable and constructive suggestions for revision. We have followed all suggestions as specified in the point-by-point responses below.

Minor comments for the text:

Abstract – ILG – the lack of shortcut extension.

We have added the elaboration of ILG as an intralingual ganglion (page number 1, line 28)

Introduction – Line 38-39. The sentence about HAP1 is not fully understandable – assumption that HAP1 interact intentionally to protect against apoptosis. Moreover, polyglutamine domain exist both in normal and mutant huntingtin. It should be specified.

HAP1 was initially recognized for its binding to polyglutamine huntingtin (Htt), the protein responsible for Huntington’s disease. This binding is enhanced by an expanded polyglutamine repeat, the length of which is also known to correlate with the age of disease onset. According to the suggestion of the reviewer, we corrected the sentences about HAP1 as “Huntingtin-associated protein 1 (HAP1) was initially identified for its binding to polyglutamine (polyQ)-expanded huntingtin, the causative protein for Huntington’s disease. HAP1 is considered protective against apoptosis or cell death induced by huntingtin with an expanded polyQ sequence (Revised text, Page 1, lines 37-40).

Reviewer 2 Report

- Immunohistochemical studies remain very interesting approaches to highlight the cellular and tissue distribution of the targets we are trying to study. However, in this work, the authors make functional extrapolations from the immunohistochemical results that are not justified.

For exemple, the authors showed a certain percentage of thigh ChAT, NOS, VIP, SP, or CGRP-ir neurons coexpressed with HAP1-ir neurons in the ILG and LRG, although I don't see what would allow the hypothesis that this could probably reflect the involvement of HAP1′ in the modulation of intrinsic sensory and parasympathetic information under particular physiological conditions in the tongue, insofar as no functional investigation was carried out.

The whole discussion is built from this point of view, which seems to me to be more of a speculation than a functional investigation necessary to venture in this direction.

The authors should be extremely prudent about functional aspects that have not been investigated accordingly.

Author Response

Immunohistochemical studies remain very interesting approaches to highlight the cellular and tissue distribution of the targets we are trying to study.

However, in this work, the authors make functional extrapolations from the immunohistochemical results that are not justified.

For example, the authors showed a certain percentage of thigh ChAT, NOS, VIP, SP, or CGRP-ir neurons coexpressed with HAP1-ir neurons in the ILG and LRG, although I don't see what would allow the hypothesis that this could probably reflect the involvement of HAP1′ in the modulation of intrinsic sensory and parasympathetic information under particular physiological conditions in the tongue, insofar as no functional investigation was carried out.

The whole discussion is built from this point of view, which seems to me to be more of a speculation than a functional investigation necessary to venture in this direction.

The authors should be extremely prudent about functional aspects that have not been investigated accordingly.

We are grateful to the reviewer for the helpful reading of our manuscript and his/her valuable feedback.

According to the suggestion of the reviewer, we corrected our discussion with a prudent approach. We updated the pointed discussion as “It is intriguing to speculate that high coexpression rate of HAP1 with different neurochemical markers in lingual ganglia probably reflect HAP1′s involvement in modulating intrinsic sensory and parasympathetic information under particular physiological conditions in the tongue”. (Indicated in red, page 10, lines 296-298).

We also agree with the reviewer that our current study clarifies the immunohistochemical distribution of HAP1; however, detailed functional investigation is necessary to explain the effects of HAP1 in the lingual ganglia. Indeed, as a limitation of our current study, we have mentioned in our original submission that “However, detailed physiological and morphological experiments need to be performed in the future to elucidate how HAP1 can regulate the parasympathetic and sensory activities in the tongue.” (Indicated in red, page 10, line 305-307).

Accordingly, we also edited the Abstract in the revised version. We replaced “This suggests that HAP1 may play a vital role in regulating or protecting parasympathetic and sensory functions in the lingual ganglia” with “The current results might serve as an essential base for future studies to elucidate the physiological/pathological functions of HAP1 in the lingual ganglia” (Indicated in red, page 1, line 31-33).

Round 2

Reviewer 2 Report

Minor comments:
- Line 305 :  

Given that physiological and morphological experiments are needed to elucidate how HAP1 may regulate parasympathetic and sensory activities of the tongue, one cannot anticipate what role HAP1 may play, so it would be preferable to indicate that the high expression of HAP1 in the ILG or LRG may suggest the involvement of HAP1 in regulating parasympathetic and sensory activities of the tongue.
The word vital is not relevant.

- Line 102 : Table 1. Details of primary antibodies used in the current study.

Table 1 is not easy to read, the separation between the columns is not clear. It would be preferable to reduce the font size and to reorganize the alignment to facilitate the reading of the table.

Author Response

Minor comments:

Point 1: - Line 305 :  Given that physiological and morphological experiments are needed to elucidate how HAP1 may regulate parasympathetic and sensory activities of the tongue, one cannot anticipate what role HAP1 may play, so it would be preferable to indicate that the high expression of HAP1 in the ILG or LRG may suggest the involvement of HAP1 in regulating parasympathetic and sensory activities of the tongue. The word vital is not relevant.

Response 1: We thank the reviewer again for carefully reading our manuscript and for the valuable suggestions for revision. We have corrected the sentence accordingly (line 305, indicated in red)

Point 2: - Line 102: Table 1. Details of primary antibodies used in the current study. Table 1 is not easy to read, the separation between the columns is not clear. It would be preferable to reduce the font size and to reorganize the alignment to facilitate the reading of the table.

Response 2: According to the reviewer's suggestion, we reduced the text font size and reorganized the alignment of Table 1.

We hope that this revised manuscript will address all of your concerns.